# Hidden hearing loss selectively impairs neural adaptation to loud sound environments

Warren Michael Henry Bakay [1,2], Lucy Anne Anderson [1], Jose Alberto Garcia-Lazaro[1], David McAlpine[1,3] & Roland Schaette [1]

Exposure to even a single episode of loud noise can damage synapses between cochlear hair cells and auditory nerve fibres, causing hidden hearing loss (HHL) that is not detected by audiometry. Here we investigate the effects of noise-induced HHL on functional hearing by measuring the ability of neurons in the auditory midbrain of mice to adapt to sound environments containing quiet and loud periods. Neurons from noise-exposed mice show less capacity for adaptation to loud environments, convey less information about sound intensity in those environments, and adaptation to the longer-term statistical structure of fluctuating sound environments is impaired. Adaptation comprises a cascade of both threshold and gain adaptation. Although noise exposure only impairs threshold adaptation directly, the preserved function of gain adaptation surprisingly aggravates coding deficits for loud environments. These deficits might help to understand why many individuals with seemingly normal hearing struggle to follow a conversation in background noise.

[1] UCL Ear Institute, 332 Gray's Inn Road, London WC1X 8EE, UK. [2] Present address: Manchester Centre for Audiology and Deafness (ManCAD), A3.16, University of Manchester, Ellen Wilkinson Building, Manchester M13 9PL, UK. [3] Present address: Department of Linguistics, The Australian Hearing Hub, Macquarie University, 16 University Avenue, Sydney, NSW 2109, Australia. These authors contributed equally: Warren Bakay, Lucy Anderson. Correspondence and requests for materials should be addressed to R.S. (email: r.schaette@ucl.ac.uk)

Detecting and understanding the meaning of sounds in difficult listening conditions, for example loud background noise, is critical to the survival of many species, and an important factor in human communication. Problems listening in background noise are often associated with hearing loss, but many people with normal hearing thresholds also struggle to hear in difficult listening environments[1], particularly so if they have a history of noise exposure[2,3] or tinnitus[4]. These, often anecdotal, reports of problems listening in background noise are consistent with increasing evidence that exposure to even a single dose of high-intensity noise can generate a substantial degree of cochlear synaptopathy, that is, a loss of, or damage to, synaptic contacts between cochlear hair cells and auditory nerve fibres (ANFs), without affecting hearing thresholds[5,6], and that ANFs with high thresholds appear to be particularly vulnerable[7]. Referred to colloquially as hidden hearing loss (HHL; ref. [8]), cochlear synaptopathy might contribute to hearing problems such as tinnitus[8–10], hyperacusis[11] and impair the ability to understand speech in loud background noise[3,12].

Loud background noise has long been thought to pose a considerable problem for neural coding of speech;[13–15] although the mechanical response of the cochlea spans the approximately 120 decibels (dB) range of sound intensities encountered in the environment, the vast majority of ANFs show low thresholds and a limited narrow dynamic range of around 10–20 dB[7,16]. Neural responses must not be driven to saturation if they are to be informative about even moderately loud sound environments, nor must the saturated responses of low-threshold fibres overwhelm the responses of the much smaller number of high-threshold fibres. One means by which the auditory brain appears to overcome this problem is through adaptive coding—neurons adapt their response functions such that their neural dynamic range shifts to encompass the most commonly occurring sound intensities in an environment. This process of rapid neural adaption, initiated in the auditory nerve[17], and amplified by midbrain[18,19] and cortical[20] processing, can double the amount of information concerning variations in sound intensity[18], and suggests a significant benefit for understanding speech in high levels of background noise. Two types of adaptation appear to be at play here: threshold adaptation, in which neural responses shift to accommodate the distribution of sound intensities in the environment, and gain adaptation, where neural responses are either amplified or attenuated.

Here, we investigate the effect of noise-induced HHL on the ability of neurons in the inferior colliculus (IC) to adapt their responses to repeated switches between relatively quiet and relatively loud sound environments. Compared to control mice, noise-exposed mice with evidence of cochlear synaptopathy show significant impairment in adaptive coding for loud environments, where neural thresholds adapt less, supra-threshold firing rates are lower and responses are less informative about the intensity distribution. We identify two specific mechanisms underlying adaptive coding in the IC—threshold adaptation and gain adaptation—which, combined, generate a stable long-term firing rate regardless of the overall sound intensities in the environment. Only threshold adaptation is directly impaired by noise exposure, leaving gain adaptation fully functional, but surprisingly the sparing of gain adaptation aggravates coding deficits in loud environments. Our data suggest that optimal adaptation performance in the central auditory system relies on access to inputs from a full population of ANFs—that is, those with low, medium and high response thresholds—with neurons adapting their response threshold and/or gain to shift their inputs from low-threshold to high-threshold ANFs as sound level increases, and that HHL might generate substantial auditory processing deficits in loud environments that cannot be compensated by neural plasticity.

## Results

**HHL and evidence of elevated central gain following exposure to noise.** We first assessed the impact of noise exposure on hearing thresholds of mice exposed to a 100 dB sound pressure level (SPL), octave-wide band of noise (8–16 kHz). One day after exposure, mice showed significantly elevated thresholds (assessed from auditory brainstem responses, ABRs, to tonal stimulation) at 11, 16, 24, and 32 kHz (t-test, $p = 0.0018$, 0.003, 0.0062, 0.003, respectively). Some 4 weeks later, however, although thresholds had recovered to pre-exposure levels (Fig. 1a), supra-threshold amplitudes of wave I of the ABR (in response to 50-µs clicks) remained low and showed significantly shallower growth with sound intensity, compared to pre-exposure values (repeated measures analysis of variance (ANOVA) for pre-exposure and 4-week post-exposure data showed a significant effect of group, $p = 0.043$, and a highly significant interaction between time of measurement and sound intensity, $p = 0.0001$; Fig. 1b). Moreover, compared to control mice 4 weeks following sham exposure, ABR wave I amplitudes were significantly smaller in exposed mice, and increased more slowly with increasing sound intensity (repeated measures ANOVA, significant effect of group, $p = 0.0072$, and highly significant interaction, $p < 0.0001$). The ABR results are therefore indicative of noise-induced cochlear synaptopathy[5]. We also assessed the effect of noise exposure on wave IV of the ABR

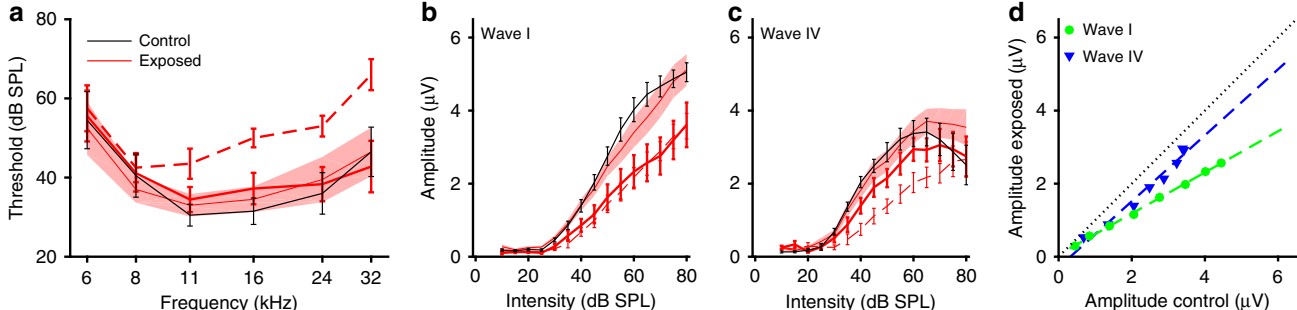

**Fig. 1** Noise-induced hidden hearing loss. ABR measurements show temporary hearing threshold loss, but permanent reduction of ABR wave I. **a** Tone-pip ABR thresholds of the left ears before (thin red line, red shaded area denotes ±s.e.m.) and 1 day (dashed red line) and 4 weeks (solid red line) after exposure to octave-band noise at 100 dB SPL ($n = 10$). Black line—thresholds of control animals ($n = 8$) 4 weeks after sham exposure. All error bars denote ±s.e.m. **b** Growth functions of ABR wave I amplitude for 50 µs clicks, colour scheme and animal numbers as in **a**. **c** Growth functions of ABR wave IV amplitude for 50 µs clicks, colour scheme and animal numbers as in **a**. **d** Plot of click-evoked ABR wave amplitudes (4 weeks after) of exposed vs. control animals shows steeper growth for wave IV than wave I, indicative of central gain increase after noise exposure

—generated by fibre tracts leading to the auditory midbrain. Wave IV showed elevated thresholds and reduced suprathreshold activity 1 day after noise exposure, but, in contrast to wave I, evidence of a much greater degree of recovery such that the amplitudes of wave IV did not differ significantly from preexposure or control values 4 weeks after noise trauma (Fig. 1c), suggesting an increase in central neural gain. We quantified this further by comparing the average amplitudes of waves I and IV in exposed animal with those of control animals for the same sound intensities (Fig. 1d). Consistent with the effects of synaptopathy, the same sound intensities evoked lower amplitudes of wave I in exposed compared to control animals (green data points fall below the line of unity in Fig. 1d), whereas amplitudes of wave IV were comparable in magnitude between exposed and control animals, particularly at higher sound intensities (blue data points in Fig. 1d). Overall, the data are consistent with an increase in compensatory neural gain in the central auditory pathways in animals with HHL, similar to those shown in animal studies[10,21,22], and reminiscent of the reported gain increase in (the equivalent human) ABR wave V in tinnitus patients with normal hearing thresholds[8,9].

**HHL impairs adaptive coding in the auditory midbrain**. The notion that elevated neural gain might generate a range of potential auditory processing deficits led us to explore potential neural markers for HHL, in particular neural markers not evident in diagnostically available measures such as the magnitude of ABR waves. To do so, we recorded sound-evoked neural activity from the auditory midbrain (IC), employing a stimulus in which the sound environment repeatedly changed between relatively quiet and relatively loud (switching stimulus). Within each environment, the range of intensities was drawn from a statistically defined distribution with 80% probability confined to a narrow range (the high probability region, HPR, Fig. 2d, right), and the remaining 20% from a broad range (24–92 dB) outside the HPR. The sound intensity within each environment changed every 50 ms, and every 7.5 s, the value of the HPR was changed, generating continuous transitions between the environments (Fig. 2d, left). HPRs were centred on 44, 56, 68 or 80 dB SPL, generating six different combinations of HPRs pairs for switching stimuli (e.g. 56–80, 44–68, etc.). In total, 91 multi-unit clusters (MUCs) from control mice, and 142 MUCs from noise-exposed mice, showed clearly defined responses (see Methods) to all HPRs in all switching stimuli and were included in all further analyses.

We first determined the capacity of IC neurons in mice to adapt to the current mean intensity of a switching stimulus. Consistent with previous studies[18,19], IC responses adapted to the current sound environment by shifting the threshold of their rate-vs.-intensity functions (RIFs) towards the HPR of that environment (Fig. 2e, f). Adapted thresholds for each HPR in each of the six possible combinations were determined by fitting broken-stick sigmoid functions to the RIFs (see Fig. 2e, f and Methods). In both control and exposed animals, there was a highly significant effect of HPR on adapted thresholds (repeated measures ANOVA, $p < 0.0001$), indicating that significant adaptive coding occurred in both groups. Nevertheless, adapted thresholds in exposed animals were generally lower than in control animals, an effect that was most evident for HPRs with higher mean sound intensities (Fig. 3). Assessed across all HPRs, adapted thresholds in noise-exposed animals (red symbols in Fig. 3a–f) increasingly lagged thresholds in controls (black symbols in Fig. 3a–f) as well as the HPR mean (i.e. the mean, background sound intensity—indicated by the coloured squares for the intensity ranges of each HPR in a switching stimulus in Fig. 3a–f) as the mean sound intensity was increased. The medians and ranges of adapted

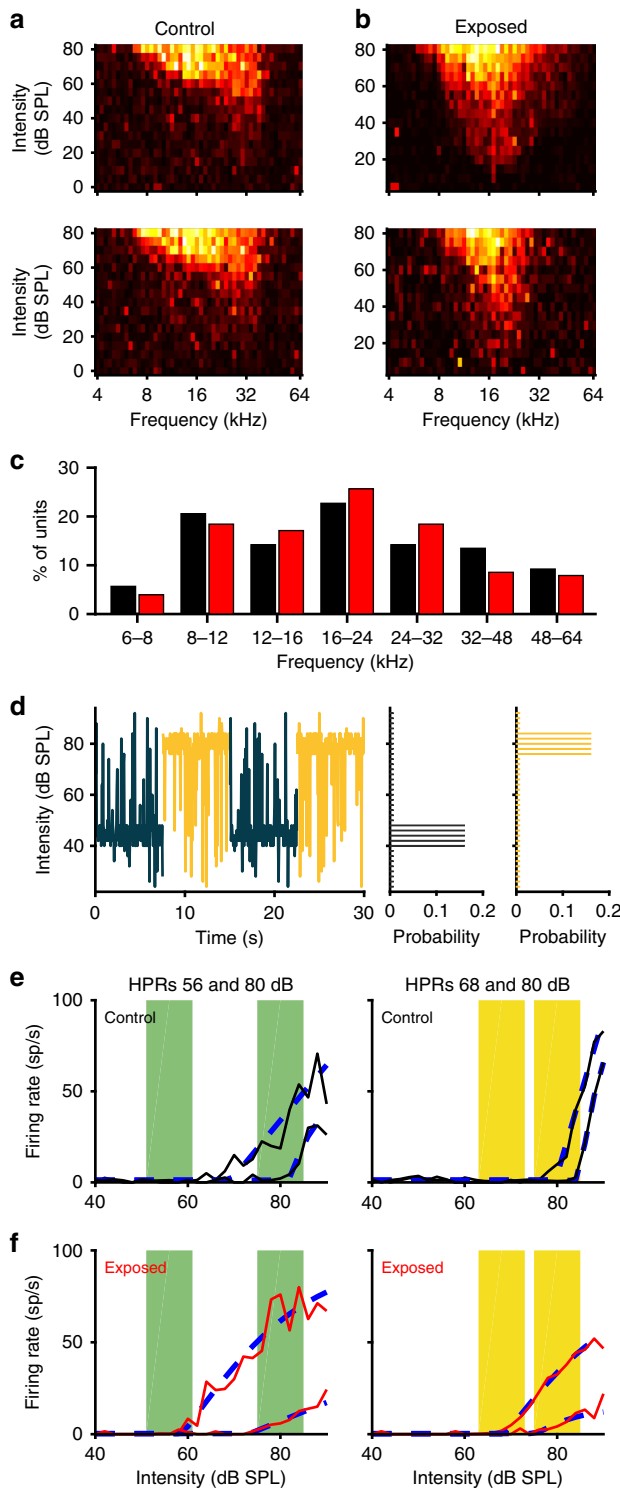

thresholds to the different combinations of HPRs are replotted in Fig. 3g, to enable a direct comparison across all HPRs. To assess whether noise exposure had a significant effect on MUC thresholds, we fitted a repeated measures model with HPR mean and context (i.e. the mean intensity of the HPR in the other half of each switching stimulus) as continuous factors (see Methods). A repeated measures ANOVA then showed highly significant effects of HPR mean, context and group ($p < 0.0001$ in all cases). Moreover, the interactions between group and HPR as well as between group and context were also highly significant ($p <$

**Fig. 2** Inferior colliculus recordings to investigate adaptive coding.
**a**, **b** Examples of frequency–response areas of IC multi-unit clusters from a control (**a**) and a noise-exposed mouse (**b**). **c** Distribution of characteristic frequencies of IC multi-unit clusters (black—control, 91 multi-unit clusters recorded from eight mice; red—noise-exposed, 142 multi-unit clusters from ten noise-exposed animals). The range of characteristic frequencies was similar in both groups, and the mean characteristic frequencies were not significantly different. **d** Illustration of the switching stimulus to investigate adaptive coding. Stimulus levels are shown in the right panel for two switch periods, and corresponding distributions of sound levels for the two HPRs (centred on 44 and 80 dB SPL) are shown on the right. **e**, **f** Examples for adapted rate-level functions (solid lines) and fits with a broken-stick hyperbolic tangent (dashed lines) for the switch stimulus with HPRs centred at 56 and 80 (left) or 68 and 80 dB SPL (right, HPRs are indicated by the coloured areas). Data from a control animal is shown in **e** and from a noise-exposed animal in **f**

0.0001 in both cases). MUCs from noise-exposed mice thus showed lower adapted thresholds, less threshold shift with increases in mean sound intensity and also a smaller effect of the stimulus context. Additionally, as some of the threshold distributions were skewed (Fig. 3), we also employed the Wilcoxon rank-sum test for pairwise comparisons between exposed and control medians for each of the 12 combinations of HPR and context. After Bonferroni adjustment for multiple comparisons, the differences were found to be significant for the 56, 68 and 80 dB HPRs for all contexts ($p < 0.05/12$ in all cases).

**Long-term adaptation to the statistical structure of sound environments is impaired by HHL.** Recently, employing an identical switching paradigm to the one we use here, Robinson et al.[23] reported a form of neural memory in the IC, in which neural responses adapt increasingly rapidly (and increasingly efficiently), to a sound environment when it is re-encountered, a process they termed meta-adaptation. Consistent with neurons coding the longer-term statistical structure of sound environments, we observed a dependence of the amount of adaptation on the mean value of both HPRs in a switching stimulus, that is, the context of the sound environment over a longer time-scale (Fig. 3). Adapted thresholds to a moderate-intensity HPR were higher when it was paired with a loud HPR (e.g. 80 dB) than when it was paired with a quiet HPR (e.g. 44 dB). This phenomenon was observed as a highly significant effect of listening context in our repeated measures ANOVA analysis across all HPRs ($p < 0.0001$, see above). However, there was also a highly significant interaction between group and context ($p < 0.0001$), demonstrating that meta-adaptation was less pronounced in noise-exposed animals; when comparing responses obtained for a quiet context to those with a loud context (e.g. 68 (44) vs. 68 (80)), MUCs from noise-exposed animals showed a smaller shift in adapted thresholds than MUCs from control animals (Fig. 3g).

Adaptive coding, as well as evidence of meta-adaptation, was also apparent in the population averages of the adapted RIFs obtained by averaging across all neural responses (Fig. 4a–h, top panels). In control animals, population RIFs showed shifts of adapted thresholds as well as their steepest slopes to within, or slightly above, the intensity range of the HPR for all HPRs, up to the highest level assessed. Moreover, an additional aspect of meta-adaptation was visible, in that the steepness of the RIFs, as well as the maximum firing rate, depended on the context in a systematic fashion for the 44, 56, and 68 dB HPRs (Fig. 4a–d). Consistently, the population RIFs obtained from switching stimuli that contained the 80 dB HPR (i.e. the loudest environment) as the

context showed the shallowest slopes and the lowest maximum firing rates.

In contrast, population RIFs obtained from noise-exposed animals (Fig. 4e–h) showed significantly lower maximum firing rates than those from controls (repeated measures ANOVA, significant effect of group, $p = 0.0087$), and there was a significant interaction between HPR and group ($p = 0.0069$); increasing the mean intensity of the HPR resulted in a greater reduction in maximum firing rates in noise-exposed than in control animals. This effect was surprising, as it might have been expected that higher, rather than lower, response magnitudes would have been observed in the noise-exposed group due to their lower adapted thresholds, especially for the louder sound environments.

To assess the extent to which adaptation influences the transmission of information about sound intensities, we calculated the Fisher information (FI) from RIFs generated by the switching stimulus (see Methods). In control animals, peak FI occurred at increasingly higher sound intensities when the intensity of the HPR was increased (Fig. 4a–d, lower panels). In contrast, although neurons in exposed animals showed FI values comparable to, or even slightly higher than those in control animals in response to the 44 and 56 dB HPRs (a potential consequence of increased neural gain at low intensities, Fig. 4e, f, lower panels), the absolute value of FI was several times lower for the 80 dB HPR in exposed animals (Fig. 4h, lower panel) than in controls (Fig. 4d, lower panel). To assess whether these differences were significant, we performed a repeated measures ANOVA on the maximum FI values (across the whole range of sound intensities) of all MUCs, with HPR and context as continuous factors, and observed a significant effect of group ($p = 0.0048$), and significant interactions between group and HPR ($p = 0.0029$), as well as group and context ($p = 0.0055$). Thus, MUCs from noise-exposed animals conveyed less information about changes in sound intensity in their responses, and also showed a stronger reduction in their capacity to transmit information when overall sound intensity was increased.

**Impairments through HHL reveal interplay between threshold and gain adaptation.** Our analysis of adapted thresholds and RIFs showed that even though neurons in exposed mice had lower thresholds than controls when adapted to the louder stimuli (especially the 80 dB HPRs), their response magnitude was lower and also decreased more strongly compared to the stimuli with a lower mean intensity. Reasoning that this might also be reflected in the overall neural response, we calculated the mean firing rate across the entire stimulus presentation for each HPR in the context of each switching stimulus. Surprisingly, in both control and noise-exposed mice, there was no overall trend for the mean rate to increase from quiet to loud environments (Fig. 5a, b), and no significant differences were observed between the groups (repeated measures ANOVA, $p > 0.05$ for both effect of group and interaction), despite large differences in mean stimulus intensities, and especially despite the deficits in threshold adaptation observed in the exposed animals, which might have been expected to generate elevated firing rates for the louder environments.

We hypothesised that this apparent stabilisation of the long-term mean firing rate could be a specific goal of adaptation in the IC, and developed a simple model to account for such an outcome. In the model, responses of IC neurons are described in terms of a broken-stick sigmoid rate-level function (see Methods), with 0 sp/s spontaneous rate and a maximum firing rate of 100 sp/s. Adapted neural threshold and the steepness of the RIF at threshold, reflecting neural response gain above threshold, are free parameters. The model illustrates that, for the HPR stimuli,

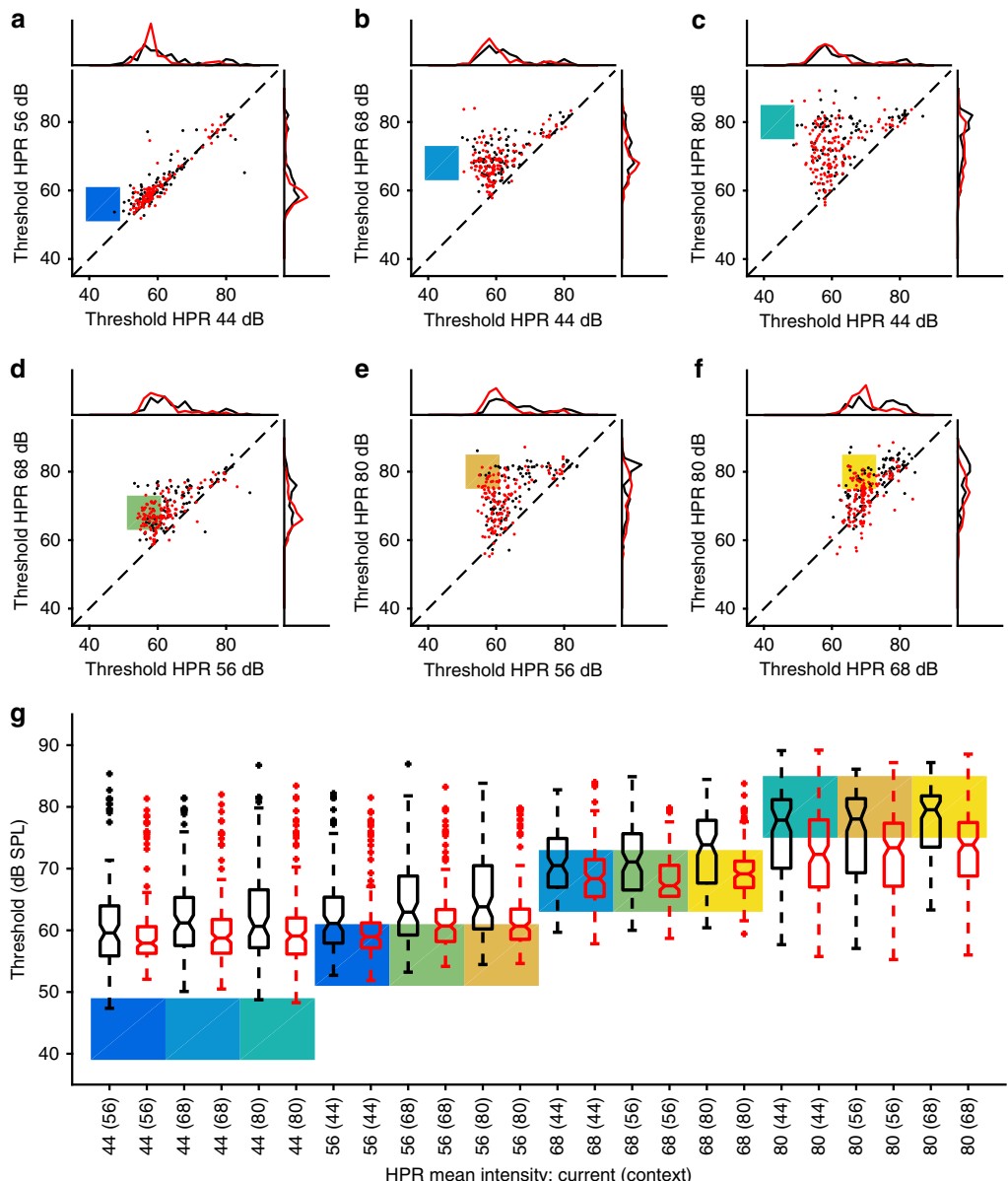

**Fig. 3** Adaptive coding, meta-adaptation and the effects of HHL. **a–f** Adapted thresholds of all multi-unit clusters (red—exposed, black—control) for all six combinations of HPRs in the switching stimulus. Coloured patches indicate the intensity ranges of the HPRs. **a** Switch between 44 and 56 dB HPRs, **b** 44 vs. 68, **c** 44 vs. 80, **d** 56 vs. 68, **e** 56 vs. 80 and **f** 68 vs. 80. **g** Box plots of adapted thresholds of RLFs from control (black) and exposed animals (red) for HPRs centred on 44, 56, 68 and 80 dB SPL. On each box, the central mark indicates the median, with the notches specifying the interval of the median, and the bottom and top edges of the box indicate the 25th and 75th percentiles, respectively. The whiskers extend to the most extreme data points not considered outliers, and the outliers are plotted individually using the + symbol. The intensity ranges of the HPRs are indicated by the coloured patches. The context, that is, the other HPR of each switching stimulus, is given in the x-axis label, and the colour of the patch refers to the panel where the corresponding raw data are shown

mean firing rate depends, first, on the adapted threshold relative to the HPR, and second, on the steepness of the RIF. When the threshold of the RIF is below the HPR, the mean rates are generally high, unless the steepness is reduced dramatically. This is because the most frequently occurring sound intensities are all above threshold in this case, and thus evoke high firing rates (Fig. 5c). Conversely, if the threshold is above the HPR, mean rates are always very low, regardless of the steepness, as sound intensities loud enough to evoke activity occur only rarely in this case (as, most commonly, sound intensity is within the intensity range of the HPR or even below). Crucially, then, if the mean rate is to be maintained at a relatively constant value, the model demonstrates a clearly defined dependence of RIF steepness, that is, supra-threshold gain, on adapted threshold (Fig. 5c, green line), with low steepness for thresholds that lie below the HPR, and systematic increases in RIF steepness as the threshold moves to higher intensities within or beyond the HPR. The resulting RIFs for a constant long-term average of the firing rate of 10 sp/s are shown in Fig. 5d.

We then investigated whether the model prediction of a dependence of gain upon adapted threshold could also be observed in our data by plotting RIF steepness as a function of

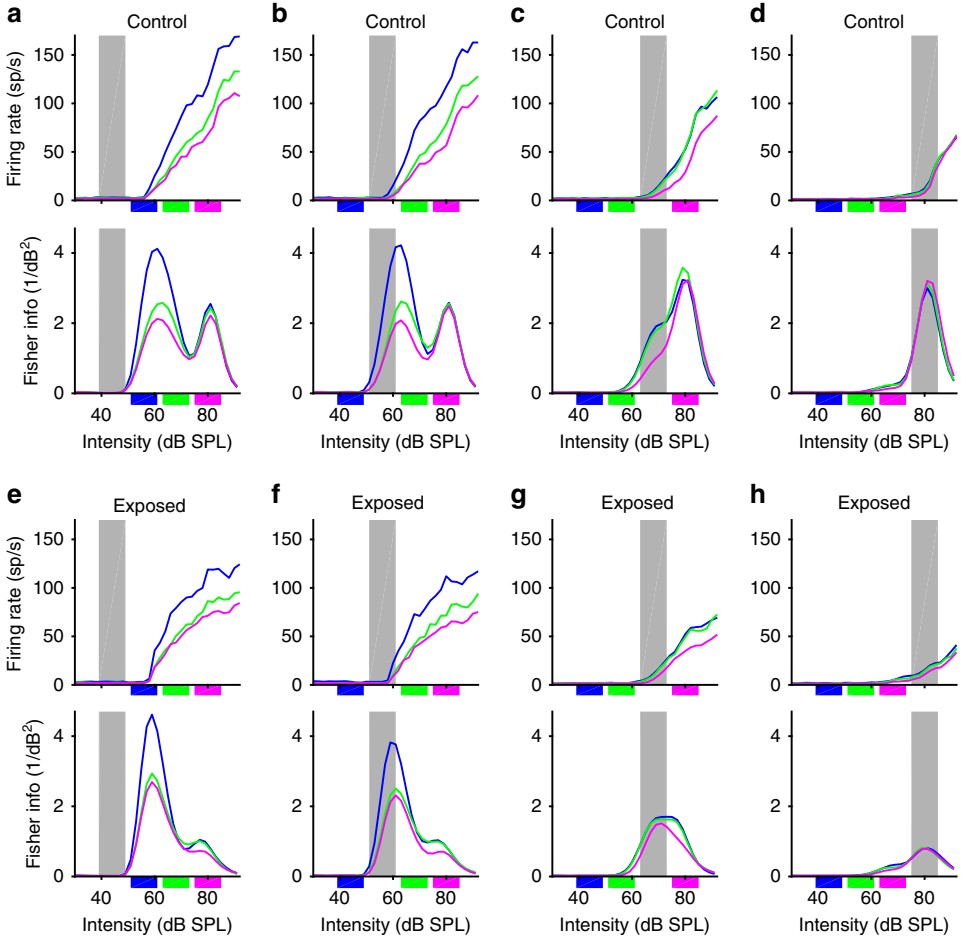

**Fig. 4** Neural population rate-vs.-intensity functions and Fisher information (FI) curves. The graphs show population averages of adapted rate-vs.-intensity functions (top panels) and average FI (bottom panels) for the different HPRs. The intensity range of the HPR for which RIFs and FIs are shown is indicated by the grey bar. The colours of the lines denote the context, that is, the other HPR within the switching stimulus, with the corresponding intensity range shown by the small coloured patches. **a–d** show data from control animals, and **e–h** show data from exposed animals

threshold (Fig. 6). For this, the steepness of the RIFs at threshold was determined from the fits with the broken-stick sigmoid function (see above and Methods). In control animals, there was a clear tendency for the steepness to be higher for recordings with thresholds above the HPR than for those with thresholds lying within or below the HPR, and this was most obvious for HPRs centred on 68 and 80 dB SPL (Fig. 6c, d). Noise-exposed mice showed comparable RIF steepness for MUCs whose adapted thresholds lay within the HPRs but, due to the relative lack of adapted thresholds above the HPR, overall fewer MUCs displayed steep RIFs for the louder HPRs (Fig. 6e–h, upper and lower panels).

We also determined the relative changes in RIF steepness in response to the different stimulus intensities by dividing the RIF steepness for each HPR by the RIF steepness obtained from the responses to the 44 (56) HPR, that is, the quietest combination of HPRs in our switching-stimulus set. For both control and noise-exposed mice (Fig. 6, middle panels), the main effect is a reduction of RIF steepness once adapted threshold falls below the HPR mean (see e.g. the middle panels of Fig. 6d, h). Additionally, for loud HPRs (68 and 80 dB SPL mean), MUCs with adapted thresholds above the HPR mean also showed an increase in RIF steepness. The main difference between control and exposed mice was the fraction of units falling in either category. The cumulative distribution functions of MUC thresholds relative to the HPR

means (Fig. 6, lower panels) showed that for the 80 dB HPR, almost all MUCs from noise-exposed animals had a threshold lower than the HPR mean (Fig. 6h, lower panel), and thus showed a reduction in their RIF steepness (Fig. 6h, middle panel), whereas in control animals, nearly half the MUCs had thresholds above the HPR mean (Fig. 6d, lower panel) and showed a concomitant increase in RIF steepness. We then carried out a correlation analysis to test whether the observed relations between threshold re HPR mean and gain change were significant. Strong and highly significant correlations were found for the 68 and 80 dB HPRS ($r$ = 0.59 to 0.75 for the 68 dB HPRs, $r$ = 0.80 to 0.87 for the 80 dB HPRs, $p < 0.001$ in all cases).

Thus, for the loudest HPRs, where the magnitude of threshold adaptation is reduced in exposed animals (Fig. 3), gain adaptation appears actively to have reduced neural responses, as would be required to maintain a relatively constant long-term firing rate. Such a reduction in response gain explains why, in noise-exposed animals, population RIFs are shallow with a low maximum firing rate for the 80 dB SPL HPRs (Fig. 4h), even though the neurons are clearly capable of responding to high sound intensities with high firing rates when these intensities are presented only rarely in the context of an otherwise quiet sound environment (Fig. 4e, f). Together, the data demonstrate that response gain is regulated in a manner dependent upon the adapted threshold, seemingly to maintain a constant long-term mean firing rate, and that this

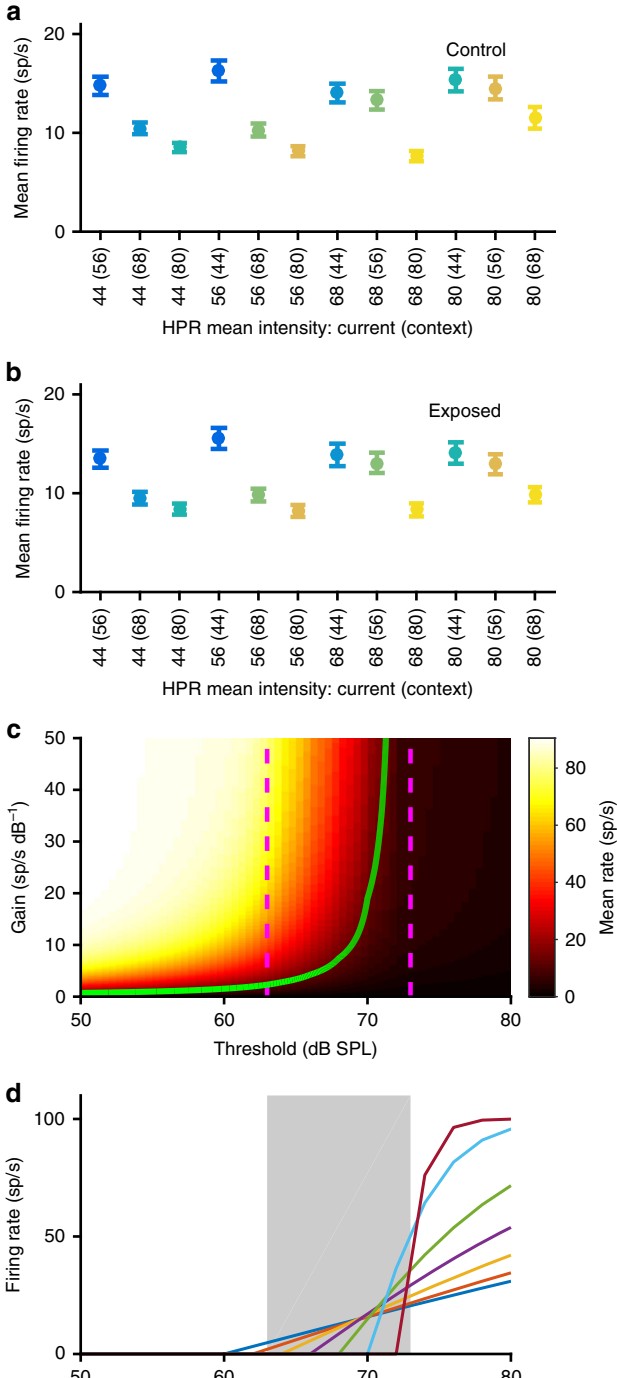

**Fig. 5** Constant mean firing rates across HPRs and adaptation model. **a** Mean firing rates over the entire duration of each HPR, averaged across all MUCs from control animals. Colour scheme for the HPRs is the same as in Fig. 3. All error bars denote ±s.e.m. **b** Mean firing rates of MUCs from noise-exposed animals. **c** Mean firing rates of a model neuron for a 68 dB HPR stimulus, in dependence upon its RIF threshold and steepness. The magenta dashed lines indicate the intensity range of the HPR, and the green line denotes those combinations of RIF threshold and gain that give a mean rate of 10 sp/s. **d** Model RIFs with different thresholds, where the steepness of each RIF has been adjusted to yield a mean firing rate of 10 sp/s for a stimulus with a 68 dB HPR

regulation seems to aggravate coding deficits originating from impaired threshold adaptation in HHL, adding insult to injury.

## Discussion

We investigated the ability of neurons in the auditory midbrain of mice to adapt their responses to sound environments that fluctuated between relatively quiet and relatively loud epochs, following exposure to noise designed to selectively damage high-threshold ANFs. Despite normal hearing thresholds, noise-exposed mice with electrophysiological evidence of cochlear synaptopathy showed specific deficits in adaptation to loud, but not quiet sound environments, with neural response functions showing smaller shifts, and neural responses carrying less information about sound level than those from control animals. The adaptation process was dissociable into threshold and gain adaptation, with only threshold adaptation impaired through noise exposure, and gain adaptation retaining normal function. However, surprisingly, the normal function of gain adaptation could not only not compensate for the threshold adaptation deficits, it actively aggravated the deficits by reducing neural responses for loud sound intensities. These results demonstrate pronounced deficits in auditory function that can occur even without apparent hearing loss, suggesting a possible explanation for difficulties listening in noise reported by many human listeners with otherwise normal hearing, and indicate the limits of compensatory plasticity in the central auditory system.

In response to the switching stimulus designed to assess adaptive coding, exposed animals showed mal-adapted responses compared to control animals. There was a significant difference in adapted thresholds between recordings from control and noise-exposed animals, which increased when the overall sound intensity was increased (Fig. 3). Moreover, for the quieter sound environments (44 and 56 dB SPL HPRs), MUCs recorded from noise-exposed mice were equally good at encoding variation in sound intensity as those recorded from control mice animals (Fig. 4a, b, e, f), whereas for the 68 and 80 dB SPL HPRs, responses recorded from exposed animals conveyed more than twofolds less information concerning variation in sound intensity (Fig. 4c, d, g, h). Finally, the effect of context (i.e. the intensity of the other half of the switching stimulus) was less pronounced in noise-exposed animals, with smaller differences in thresholds between quiet and loud contexts, compared to animals in the control group (Fig. 3). Deficits for loud environments are consistent with the reported bias in noise damage to affect high-threshold ANFs most strongly[7], indicating that input from high-threshold ANFs is crucial for the accurate neural representation of loud sound environments, as their loss cannot be compensated through neural adaptation in the central auditory system. The bimodal shape of the FI curves obtained for the 44 and 56 dB SPL HPRs from control animals (Fig. 4a, b), which is almost absent in noise-exposed animals (Fig. 4e, f), also indicates that the fraction of IC neurons with high response thresholds may be reduced after HHL, which will directly affect the encoding of loud sound environments. How exactly the responses of the different neuron types in the IC will be affected by HHL will require further investigation, likely employing single neuron recordings.

The specific deficits caused by noise exposure that generates only a temporary elevation in hearing thresholds[5] and preferentially damages high-threshold ANFs[7] enabled us to dissociate the different components contributing to adaptive coding in the mouse auditory midbrain. The interplay of threshold and gain adaptation emphasises responses of those neurons whose adapted thresholds lie at or just above the most commonly occurring sound intensities (Fig. 6), regardless of the mean intensity of the environment, thus contributing to intensity-

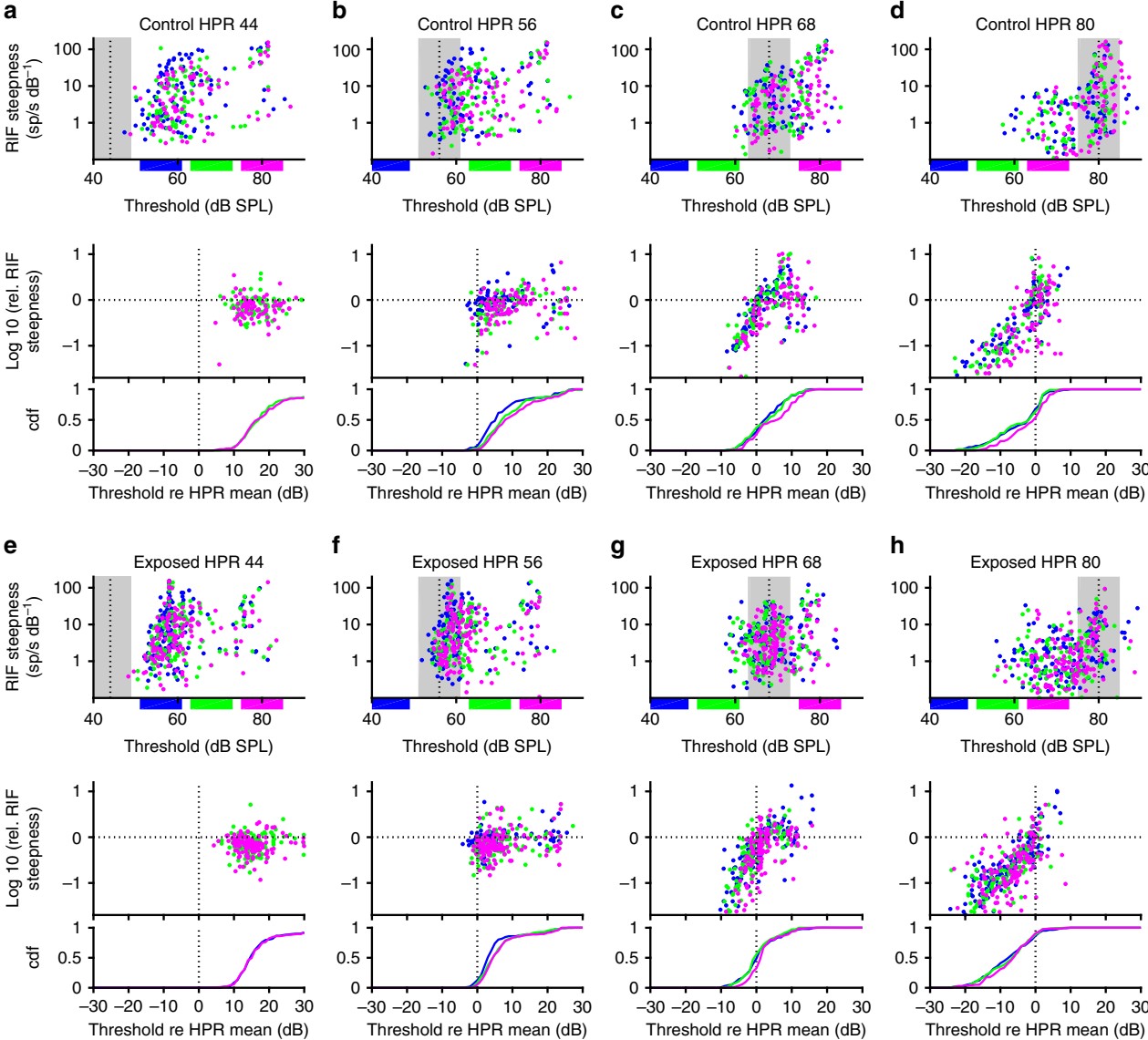

**Fig. 6** Interplay between threshold and gain adaptation. **a–d** MUCs from control animals had the highest RIF steepness when the adapted threshold was towards the upper edge or above the HPR (top row, the colours denote the intensity of the HPR during the other half of the switching stimulus, see Fig. 4) Within-unit analysis of gain changes relative to the RIF steepness for the 44 (56) HPR (middle row) demonstrated that RIF steepness is reduced when the adapted threshold falls below the HPR mean. The cumulative distribution of MUC thresholds is shown in the bottom row. **e–h** MUCs from exposed animals also showed a tendency towards steeper RLFs with higher adapted thresholds, but there were much fewer units with thresholds above the HPR range for loud HPRs (bottom row). For the 80 dB HPRs, most MUCs showed a decrease in RIF steepness relative to the RIFs in response to the 44 (56) HPR

invariant auditory coding[24]. Specifically, increases and reductions in the steepness of RIFs through modification of central gain could be employed to amplify the responses of a subset of neurons from a population with a range of response thresholds, dampening the response to background noise whilst enhancing the salience of responses to rarer stimuli emerging from this background. Such a separation of transient signals from a background through adaptation has, for example, also been reported for weakly electric fish[25], illustrating that the mechanism we describe here follows universal principles of neural information processing. Our modelling work suggests that this can be achieved through a simple objective function, where gain adaptation stabilises the mean firing rate dependent on the degree of threshold adaptation, automatically increasing the gain for neurons whose thresholds are adapted to respond to rare stimuli, and reducing it for neurons with thresholds in the range of commonly occurring sound intensities. The fact that gain adaptation reduced

neural responses to loud environments in mice with HHL, where threshold adaptation failed to keep up with increasing mean sound intensity, demonstrates that a significant component of gain adaptation must arise downstream of threshold adaptation.

Whilst a certain amount of threshold, as well as gain, adaptation occurs within individual ANFs[17,26], the greater adaptative capacity of neurons in the central auditory system[17] indicates a significant contribution of central mechanisms. In our recordings, IC responses showed threshold shifts in excess of 30 dB, consistent with previous reports, suggesting that they might adapt their responses to convey inputs from either low-threshold or high-threshold ANFs, depending on the prevailing acoustic environment. At a neuronal population level, this selection process may also be achieved through the gain adaptation mechanism that we have described, which in a loud environment helps to increase the salience of the response of units with high response thresholds by reducing the response of those with response

thresholds below the average sound intensity. Our data also indicate that the adaptation mechanisms comprise different time scales, as evident by the fast adaptation to changes in mean sound level (occurring every 7.5 s in our switching stimuli), and by slower adaptation processes enabling meta-adaptation, where the context (the other half of the switching stimulus) influences adaptation in the current sound environment, facilitating adaptation to the overall statistics of the stimulus on a longer time-scale. The potentially longer time-scale of at least some element of gain adaptation is consistent with the recent report of a cortical influence on adaptive coding by midbrain neurons[23].

Cochlear synaptopathy and neuropathy have been shown to be a part of the normal ageing process in mice[27] and humans[28]. Our data might help explain the age-related decline in the ability to understand speech in noise, only partially understood in terms of elevated hearing thresholds[29]. Moreover, age-related synaptopathy has been shown to occur at a faster rate following previous noise exposure with temporary hearing threshold shift[30], indicating that a misspent youth without hearing protection could manifest itself in an early onset of problems understanding speech in high-level background noise. Our data also reinforce the notion that audiometry, that is, testing thresholds for hearing, might not be sufficient to generate a comprehensive picture of the hearing status of an individual, since normal hearing thresholds do not guarantee the absence of cochlear damage and normal hearing performance. Elevated thresholds might be the final manifestation of an extensive cochlear damage process, as surprisingly few ANFs or even inner hair cells are necessary to generate normal hearing thresholds[22,31,32], and thus tests of functional hearing might be better suited to assess whether cochlear damage is present, e.g. for medico-legal assessment. However, it is likely that to detect HHL-related performance deficits in humans, great care will be required in test design[33], and our results show that deficits might only be encountered for specific stimulus conditions. Moreover, the noise exposure used here was specifically tailored to produce HHL, and mice might be especially prone to this pathology, as they have a relatively high proportion of high-threshold ANFs. The true extent of the problem for humans thus remains to be determined.

Finally, our findings have direct implications for rehabilitation of hearing loss, where achieving better performance for speech in noise has remained one of the toughest challenges[34,35]. It is now becoming clear that hearing threshold loss might almost always be accompanied by a considerable degree of cochlear synaptopathy[10,28,36]. In our study, functional deficits caused by synaptopathy occurred only for high sound intensities, with normal performance at lower sound intensities, suggesting the existence of an optimal sound intensity for auditory coding and thus for understanding speech in noise in hearing-impaired listeners and those with HHL. Attenuation of sound by 10–20 dB instead of amplification might then prove more effective in improving speech understanding in high levels of background noise, as this manipulation could move the signal back into the range where neural coding is normal (and more ANFs are available for encoding the signal). Such an optimal range for listening performance could be determined prior to hearing aid fitting and targeted by the hearing aid algorithm, to improve hearing function in difficult listening conditions.

## Methods

**Subjects.** Subjects were 18 male CBA/Ca mice. Mice were 7–13 weeks old at the time of noise exposure. Control animals were age-matched littermates. ABRs were recorded 1–5 days prior to, 1 day after and 4 weeks after noise or sham exposure. IC recordings took place 4 weeks after noise exposure. At the end of the final experiment, mice were overdosed with an intra-peritoneal (i.p.) injection of sodium pentobarbital. All experiments were performed in accordance with the United Kingdom Animal (Scientific Procedures) Act of 1986, under a project licence approved by the UK Home Office (PPL 70/7202).

**Anaesthesia.** All procedures were carried out under ketamine/medetomidine (i.p.) anaesthesia. Pedal reflex and breathing rate were checked every 30 min.

**Stimulus generation and delivery.** Stimuli were generated using a Tucker David Technologies (TDT) RX6 processor, attenuated as needed (TDT PA5), and amplified (TDT SA2). For ABR and IC recordings, stimuli were presented in free-field condition with the speaker (TDT FF1) positioned at a 45° angle to the animal's axis at a distance of approximately 15 cm. The ear contralateral to the speaker was blocked using a foam earplug. Before the start of each experiment, the transfer function of the speaker was measured with a microphone (4939, Brüel and Kjær) placed at the location of the animal's ear with the animal in place. This function was used to calibrate individual tones so that the overall output of the speaker was flat across frequency to ±3 dB.

**Noise exposure.** Anaesthetised mice were positioned in a sound-proof booth on a heated pad underneath the centre of a speaker (Stage Line MHD-220N/RD) 45 cm above. Noise exposure was performed with an octave-band noise (8–16 kHz) at 100 dB SPL for 2 h, with both ears open. Control animals underwent a sham exposure protocol with the same anaesthesia and duration, where no noise was presented.

**ABR recording and analysis.** ABR recordings from anaesthetised mice were obtained using subdermal needle electrodes (Rochester Medical), one inserted at the vertex, and one each behind the ipsilateral and contralateral pinnae. Electrode signals were low-pass filtered (7.5 kHz cut-off frequency) and recorded at 24 kHz sampling rate (TDT RA4LI, RA4PA and RX5). For analysis, ABR data were filtered using a bandpass filter (100–3000 Hz). Stimuli were tone pips (5 ms total duration with 1.5 ms rise/fall time; frequencies 6, 8, 11, 16, 24 and 32 kHz) or clicks (50 µs duration), with intensities 0–80 dB SPL in 5 dB steps, delivered at a rate of 20/s. ABR thresholds were determined visually by estimating the lowest sound level at which deflections in the ABR waveform were greater than the background variability in the waveforms. Measurements of wave amplitudes were performed using custom Matlab software: a time window containing the wave of interest was selected by the user, and the software then detected maxima and minima of the ABR traces within that window. ABR wave I amplitudes were measured from the peak to the following trough. Wave IV amplitudes were measured from their peak to SN10 (the second trough after wave IV), since the trough immediately following wave IV was not reliably present, often turning into a shoulder or even vanishing completely at high sound intensities.

**Extracellular recordings in IC.** Animals were anaesthetised with ketamine/medetomidine, followed by administration of dexamethasone and atropine sulphate. Lactated Ringers solution was given every 2 h to maintain hydration. The animal's temperature was maintained at 37.5 °C using a homeothermic blanket connected to a rectal thermistor. Breathing rate was monitored throughout the surgery, and then at appropriate intervals throughout the experiment. Once the pedal reflex had been abolished, the mouse was placed in a nose clamp to stabilise the head while leaving the ears free. To access the IC, a craniotomy (circular, 1.5 mm diameter, centred ≈5.25 mm posterior to Bregma, ≈0.75 mm lateral to midline) was performed on the right-hand side, revealing the surface of the right IC. Extracellular, multi-unit recordings were made using either single-shank or double-shank silicon multi-electrodes with 16 recording sites (1 × 16 linear array with 100 µm spacing between recording sites; 4 × 4 tetrode arrays with 100 µm spacing along diagonal, 150 µm spacing across tetrode; NeuroNexus).

The probe was advanced manually until the tip just touched the collicular surface. Using a remote hydraulic microdrive (Neurocraft, FHC Inc.), the electrode array was initially advanced rapidly by 2000 µm, to minimise the duration of tissue compression during the initial penetration, and then retracted by 500 µm. Multiple penetrations were performed for each subject, at different locations and depths, to cover as much of the IC as possible. Penetrations were made through the IC, determined visually, with access to the central nucleus verified by the tonotopic gradient of frequency tuning, assessed from frequency-vs.-intensity response areas (FRAs). Electrode signals were recorded at 24 kHz sampling rate and bandpass filtered between 300 and 9000 Hz (TDT RX5).

Frequency–response areas were obtained for 100 ms tone pips from 4–70 kHz (1/8th octave steps), from 0 to 80 dB SPL in 5 dB steps. The FRA measurement was repeated three times.

To investigate adaptive coding, we used a stimulus where the intensity of a broadband noise (2–45 kHz) was changed every 50 ms. With 80% probability the intensity was chosen from an HPR, and with 20% probability from the remaining intensities of the 24–92 dB SPL range. We used four different HPRs centred on 44, 56, 68 and 80 dB SPL. For each "switching stimulus", two HPRs were switched every 7.5 s, for a total duration of 300 s (see also Fig. 2c).

**Analysis of IC data**. Voltage traces were bandpass filtered between 500 Hz and 5 kHz to reduce noise and local field potentials. Electrical events were classified as neural spikes using root mean square thresholding, with the threshold set to four times the root mean-squared value. This procedure was used for both tetrode electrodes and linear arrays. Spike sorting using KlustaKwik was attempted on data from tetrode arrays, but did not yield evidence of single units. We therefore chose to derive multi-unit responses from tetrode electrodes through thresholding. For each tetrode, characteristic frequencies, thresholds and rate-level functions of the responses from each electrode were inspected visually, and typically one MUC was selected for further analysis. In rare cases, where the CFs differed by more than 0.5 octaves between two electrodes, 2 MUCs were selected from a single tetrode.

Rate-level functions were derived from the responses to the adaptive coding stimulus by counting spike events during the 50 ms epochs, with spike times adjusted for IC latency, and then determining the average spiking response for each intensity. MUCs with a maximum firing rate of <10 spikes/s in response to the adaptive coding stimulus were excluded from further analysis. To determine the thresholds of the adapted rate-level functions, broken-stick sigmoid functions based on a hyperbolic tangent were fitted using a least-squares fitting algorithm, with the threshold $I_{th}$, the spontaneous rate $r_{sp}$, the maximum rate $r_{max}$ and the steepness $k$ as fit parameters:

$$r(I) = r_{sp} + (r_{max} - r_{sp}) \times \tanh\left[(I - I_{th}) \times k\right]_{+} \tag{1}$$

All fits were inspected visually, see Fig. 2d, e for example results. Only MUCs with clearly defined response thresholds and good RIF fits for all 12 HPRs in the six switching stimuli were included in further analyses.

**Fisher information**. We used an approximation formula to estimate the FI for individual multi-units from their rate-level functions[18]:

$$f_a(I) = r_a'(I)^2 / \sigma_a^2(I), \tag{2}$$

where $r_a'(I)$ is the differential of the rate-level function after smoothing with a Gaussian filter with a standard deviation of 4 dB. The differential was determined by calculating the steepness of the curve between two measured intensities. Furthermore, we assumed that the variance of the neural response $\sigma_a^2(I)$ was equal to the mean.

**Adaptation model**. Neuronal responses to broadband noise were represented through a sigmoid rate-level function $r(I)$, as also used in the analysis of IC data (see above). The maximum firing rate $r_{max}$ was set to 100 sp/s, close to the average maximum firing rate of MUCs for the HPRs centred on 44 dB SPL (Fig. 4), and the spontaneous rate $r_{sp}$ to 0 sp/s. The threshold $I_{th}$ and the steepness $k$ were free parameters. With the known probability $p_I(I)$ of occurrence of individual sound intensities $I$ in our switching stimuli, the mean firing rate of the model neuron can then be simply calculated as

$$\langle r \rangle = \sum_I p_I(I) r(I), \tag{3}$$

to determine the dependence of the mean rate on response threshold and gain.

All data analyses and modelling were done using MATLAB (The MathWorks Inc., Natick, MA, USA). To test for significant differences, $t$-tests, Wilcoxon rank sum tests, and repeated measures ANOVAs (Matlab functions fitrm and ranova) were used. All error bars are ±s.e.m.

## Data availability

The data that support the findings of this study are available from the corresponding author on reasonable request.

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

## Acknowledgements

We would like to thank Bjorn Christianson for technical help during the implementation of the adaptive coding stimulus. This study was supported by Action on Hearing Loss (Studentship S25 and International Project Grant G80) and the Medical Research Council UK (grant MR/L022311/1).

## Author contributions

W.B., D.M. and R.S. designed research; W.B. and L.A. performed experiments; W.B., J.A. G.-L., D.M. and R.S. analysed data; R.S. prepared figures; W.B., D.M. and R.S. wrote the manuscript.

## Additional information

**Competing interests:** The authors declare no competing interests.

