## [Peer Review File · Nature Communications]

Reviewers' Comments:

Reviewer #1:

Remarks to the Author:

The authors examined the effects of hidden hearing loss following temporary threshold shift on intensity adaptation. They found that midbrain neurons maintain mean firing rates presumably by adjusting gain according to the adaptive threshold. In animals with hidden hearing loss, the adaptive threshold shift is impaired at the high intensity range, but the gain control is intact so that the mean firing rate is maintained. The authors further showed that the neurons in animals with hidden hearing loss provided less information on sound intensity in a loud environment, which has implications on difficulties of hearing in noise that is well documented for people with hidden hearing loss. These findings are consistent with the impaired high-threshold auditory nerve fiber responses, and, in hindsight, are not surprising. But they had never been documented so clearly before. The dichotomy/dissociation between threshold adaptation and gain control is particularly insightful. The findings are interesting and timely. The experiments are well carried out and the results are well presented. I only have a few minor points that need to be corrected.

- 1) Although the mean firing rate was maintained across different sound intensity levels, it was overall lower for the exposed group than the control groups. This should be interpreted in the context of the threshold-gain model.
- 2) Figure 2 panel labeling is inconsistent with the text in the legend.
- 3) Figure 2d HPRs should be 44dB and 80dB, but the legend says 56dB.
- 4) Figure 5d, the area size in the HPR under each curve should be the same. But one of the curve (the one with highest threshold) is clearly different.

Reviewer #2:

Remarks to the Author:

In this paper, the authors show that in a mouse model of 'hidden hearing loss', the ability of neurons in the inferior colliculus to adapt to loud levels of noise is degraded. The authors argue that neurons show lower threshold adaptation, but intact gain adaptation, that reduces the amount of information these neurons can carry about the stimulus in loud noise levels. This is consistent with hearing-in-noise difficulties experienced by humans with hidden hearing loss. The authors therefore conclude that they have found a correlate for these deficits in the inferior colliculus.

Hidden hearing loss is a fairly recently-discovered phenomenon, and many facets of this condition are poorly understood. This is a unique condition where sensory abilities are apparently preserved (using traditional diagnostic tests), but the destruction of a specific population of synapses that might carry unique information results in a deficit that causes serious quality-of-life issues. Therefore, this is quite an interesting and timely study that will interest a wide range of neuroscientists, clinicians and audiologists. This paper is certainly suitable for Nature communications.

This study attempts to directly tackle a central question in our understanding of this disorder. The experiments in this study adapt a clever paradigm to study the encoding ability of neurons in noise. The experiments and most of the analyses are carried out well, and the paper is well-written.

However, enthusiasm for the study is dampened by several concerns. The paper stops short of suggesting possible mechanisms that could explain these deficits. There are concerns about the quantification and statistical analysis of a central claim. In interpreting the data, alternative hypotheses are not considered. It is somewhat disappointing that the authors perform multiunit recordings from anesthetized animals, which for electrophysiology, is not state-of-the-art. Therefore,

more analyses and possibly more data are necessary for this paper to move to publication.

Major concerns:

1) The central claim of the paper is that after NE, neurons in animals with putative hidden hearing loss show 1) lower threshold adaptation, whereas 2) gain adaptation is preserved. The data supporting this central claim are presented in Figures 3 and 4 of the manuscript. The argument that gain adaptation is preserved is contingent upon observing lower threshold adaptation (as modeled in Fig. 5). Therefore, Fig. 3 is crucial to support the central claim of the paper.

In Fig. 3, I can see from the scatter plots that there is a trend towards lower threshold adaptation in NE animals. However, the statistical analysis to show that this is a significant effect is *not* appropriate, and the effect size is not quantified. It appears that the authors have simply conducted a number of t-tests comparing the threshold between control and NE animals for each current(context) combination. This raises two issues: 1) it assumes that the underlying data are normally distributed, and 2) it raises a multiple comparisons problem.

Regarding (1), from the marginal histograms in Fig. 3d-f, the data may not normally distributed. The use of a Tukey plot in 3g (rather than mean/SD) suggests that the authors recognize this, and want to compare medians. Therefore, non-parametric statistical tests, rather than t-tests, should be used. But (2) is a bigger concern.

To deal with (2), for normally distributed data I would suggest an ANCOVA with control and NE as groups, and HPR mean intensity as a covariate (setting aside meta-adaptation for now). The authors could try this and test for normality of the residuals. It is possible that this approach is sufficient. But if data are strongly non-normal, it is likely that a more sophisticated approach (such as a generalized linear model) is necessary to test for statistically significant differences between control and NE.

This would also allow the authors to estimate the magnitude of the difference in 'threshold shift' between control and NE, which the authors do not present in the current manuscript. An estimate of this magnitude is necessary to evaluate the model in Figure 5. It appears that a threshold shift difference of ~12dB is necessary to obtain the difference between control and NE responses according to the model. Yet, judging from Fig. 3g, the actual threshold shift difference between control and NE is ~5dB. This would argue against gain adaptation being preserved.

I do not think these concerns are just statistical details. As presented, it appears that both central claims of the paper rest on a significant threshold shift difference. I would therefore encourage the authors to quantitatively nail this down.

2) A central conceptual concern is how the authors interpret "gain adaptation". The authors do not clearly define this term in the paper. I strongly suggest they do so in the Introduction. The closest they come to defining it is in Line 396 – gain adaptation is necessary to maintain a stable long-term firing rate. If the claim that noise exposed animals (NE) have lower threshold adaptation is true, then by having intact gain adaptation, there is lower information about sounds at high prevailing noise levels. However the question arises whether the preservation of long-term firing rate is an intended result or an epiphenomenon. An alternate explanation may be that the function of 'gain adaptation' (or alternate terminology) is to preserve the dynamic range of neural responses. At high thresholds, high gain ensures that the entire range of neural response rates is used for encoding. In this interpretation, it is a *failure* of 'gain adaptation' that leads to a loss of dynamic range in high noise levels. This decouples 'gain adaptation' from threshold adaptation. Given the issues with threshold adaptation (above), this might be an attractive alternative hypothesis to consider and discuss. The point here is

that there may be a more straightforward explanation for what the authors observe.

3) Related to above, the authors do not comment on the two-peaked FI curves observed in the control condition (Fig. 4a), but is conspicuously absent in the NE condition. Does the second peak reflect the effect of high-threshold fiber inputs? If so, are the observed differences between control and NE basically the presence or absence of high-threshold inputs? If this is the case, one role of high-threshold inputs might be to preserve dynamic range at loud levels.

4) Is gain adaptation really preserved? Although the mean spike rates in NE is roughly constant over HPRs, the magnitude is reduced significantly (as the authors point out). In the model, then, the constant-rate curve will be shifted to the right. Is this not a change in gain adaptation?

5) Did the authors attempt any spike sorting on their data, given that they used tetrodes? Are single-unit data consistent with the multi-unit data that are presented? While they observe population-level effects from the multi-unit data, it is disappointing that they do not mention how single units behave because this could provide insight into underlying mechanisms. For example, the observed two-peaked FI curves could arise from distinct single neurons, which could carry implications for how high-threshold fibers affect AN activity. Further analyses comparing sorted single-unit data to the multi-unit data presented would significantly strengthen the paper.

Minor concerns:

1) The authors mention that they obtain FRAs. Could they comment on whether there were differences between control and NE FRAs at high sound levels?

2) Please provide details on the number of units recorded, inclusion criteria for analysis, and the number of units analyzed. Since recordings are from a tetrode array, and it seems that each channel was treated independently, how do you ensure that units are not replicated across channels in a given tetrode? Some of this information is in a figure legend, but it belongs in the Methods section.

3) Were these units recorded from the core or shell of IC? This is an important distinction and should be stated in Methods.

L78: "... suprathreshold firing rates were lower", but L310 "...adapted firing rates were slightly higher than in control animals for the quieter HPRs". I think L78 refers to the overall mean, and not all suprathreshold firing. Please clarify.

L213: Is the t-test paired? Were multiple t-tests run to test whether 1-day post and pre were different at each frequency? If so, there is a multiple comparisons issue. Possibly, the 48kHz data point will not survive multiple comparisons correction. Since the authors do have many more NE neurons at 48-64 Khz, I wonder if they are underestimating their effects by including neurons that were in a frequency range not affected by NE. It might be worthwhile to drop the 48-64KHz neurons on this basis.

L220: 'more slowly' → 'at a slower rate'

L223: Any anatomy to confirm synaptopathy with your noise exposure paradigm would be nice.

L520: This is an intriguing suggestion, and possibly easily tested using something as simple as earplugs. Is there any published data testing this that might support your suggestion?

First, we would like to thank both reviewers for their detailed assessment of our manuscript and their constructive feedback. Following their suggestion, we undertook a series of additional experiments, have added more data, revised the statistical analysis, and edited the manuscript. We hope that the revised version will now be suitable for publication.

Our detailed responses to the reviewers' comments can be found below. Reviewers' comments are in italics, our replies in normal typeface.

Reviewer 1:

The authors examined the effects of hidden hearing loss following temporary threshold shift on intensity adaptation. They found that midbrain neurons maintain mean firing rates presumably by adjusting gain according to the adaptive threshold. In animals with hidden hearing loss, the adaptive threshold shift is impaired at the high intensity range, but the gain control is intact so that the mean firing rate is maintained. The authors further showed that the neurons in animals with hidden hearing loss provided less information on sound intensity in a loud environment, which has implications on difficulties of hearing in noise that is well documented for people with hidden hearing loss. These findings are consistent with the impaired high-threshold auditory nerve fiber responses, and, in hindsight, are not surprising. But they had never been documented so clearly before. The dichotomy/dissociation between threshold adaptation and gain control is particularly insightful. The findings are interesting and timely. The experiments are well carried out and the results are well presented. I only have a few minor points that need to be corrected.

We would like to thank the reviewer for this positive assessment of our work!

1) Although the mean firing rate was maintained across different sound intensity levels, it was overall lower for the exposed group than the control groups. This should be interpreted in the context of the threshold-gain model.

The main reason for the difference in mean firing rates was a difference in average subthreshold (spontaneous) firing rates between exposed and control animals, not of the suprathreshold responses. However, upon applying stricter criteria for the inclusion of MUCs into our analysis of the larger data set, this difference is no longer significant (see revised figure 5), suggesting this was a spurious effect, and is not evident with a larger data set.

2) Figure 2 panel labeling is inconsistent with the text in the legend.

Thanks for spotting this mistake, the legend has been amended.

3) Figure 2d HPRs should be 44dB and 80dB, but the legend says 56dB.

Thanks for spotting this mistake, it has been corrected.

4) Figure 5d, the area size in the HPR under each curve should be the same. But one of the curve (the one with highest threshold) is clearly different.

In the model, we have imposed a maximum value for the steepness of RIFs, to account for biological limitations. This maximum had been reached for the curve with the highest threshold, and thus the area under this RIF was indeed smaller than the areas under the other ones. This explanation was lost in editing when we shortened the paper. In the end, we have decided simply to remove this curve, in order to not create confusion. Thanks for pointing this out.

Reviewer 2

In this paper, the authors show that in a mouse model of 'hidden hearing loss', the ability of neurons in the inferior colliculus to adapt to loud levels of noise is degraded. The authors argue that neurons show lower threshold adaptation, but intact gain adaptation, that reduces the amount of information these neurons can carry about the stimulus in loud noise levels. This is consistent with hearing-in-noise difficulties experienced by humans with hidden hearing loss. The authors therefore conclude that they have found a correlate for these deficits in the inferior colliculus.

Hidden hearing loss is a fairly recently-discovered phenomenon, and many facets of this condition are poorly understood. This is a unique condition where sensory abilities are apparently preserved (using traditional diagnostic tests), but the destruction of a specific population of synapses that might carry unique information results in a deficit that causes serious quality-of-life issues. Therefore, this is quite an interesting and timely study that will interest a wide range of neuroscientists, clinicians and audiologists. This paper is certainly suitable for Nature communications.

This study attempts to directly tackle a central question in our understanding of this disorder. The experiments in this study adapt a clever paradigm to study the encoding ability of neurons in noise. The experiments and most of the analyses are carried out well, and the paper is well-written.

However, enthusiasm for the study is dampened by several concerns. The paper stops short of suggesting possible mechanisms that could explain these deficits. There are concerns about the quantification and statistical analysis of a central claim. In interpreting the data, alternative hypotheses are not considered. It is somewhat disappointing that the authors perform multiunit recordings from anesthetized animals, which for electrophysiology, is not state-of-the-art. Therefore, more analyses and possibly more data are necessary for this paper to move to publication.

We would like to thank the reviewer for the detailed and generally positive assessment of our work. As requested, we have performed additional experiments, adding data from an additional four control and five noise-exposed mice to the data set, and completely re-done the statistical analysis. As data from the additional experiments substantially increased the number of recordings available for analysis, we could apply stricter selection criteria to the multi-unit clusters that were included in the analysis, using only those that had a clearly identifiable response threshold in all stimulus conditions. This has then enabled us to employ a more rigorous approach to the statistical analysis. As the number of recordings is now the same across all stimulus conditions, we were able to perform a repeated measures analysis across all conditions. Due to the increase in the size of the data set in combination with a more powerful analysis, we could corroborate the significance of all effects.

Major concerns:

1) The central claim of the paper is that after NE, neurons in animals with putative hidden hearing loss show 1) lower threshold adaptation, whereas 2) gain adaptation is preserved. The data supporting this central claim are presented in Figures 3 and 4 of the manuscript. The argument that gain adaptation is preserved is contingent upon observing lower threshold adaptation (as modeled in Fig. 5). Therefore, Fig. 3 is crucial to support the central claim of the paper.

*In Fig. 3, I can see from the scatter plots that there is a trend towards lower threshold adaption in NE animals. However, the statistical analysis to show that this is a significant effect is *not* appropriate, and the effect size is not quantified. It appears that the authors have simply conducted a number of t-tests comparing the threshold between control and NE animals for each current(context) combination. This raises two issues: 1) it assumes that the underlying data are normally distributed, and 2) it raises a multiple comparisons problem.*

Regarding (1), from the marginal histograms in Fig. 3d-f, the data may not normally distributed. The use of a Tukey plot in 3g (rather than mean/SD) suggests that the authors recognize this, and want to compare medians. Therefore, non-parametric statistical tests, rather than t-tests, should be used. But (2) is a bigger concern.

Our reason for using box-whisker plots was to provide a different visualisation of the data in addition to the scatter plots, and to enable easy comparisons across the different stimulus conditions.

To deal with (2), for normally distributed data I would suggest an ANCOVA with control and NE as groups, and HPR mean intensity as a covariate (setting aside meta-adaptation for now). The authors could try this and test for normality of the residuals. It is possible that this approach is sufficient. But if data are strongly non-normal, it is likely that a more sophisticated approach (such as a generalized linear model) is necessary to test for statistically significant differences between control and NE.

We would like to thank the reviewer for the detailed assessment of our statistical analysis and for the constructive suggestions on how to improve it. In our revised analysis, we now fit a repeated measures model to the data, with HPR and Context as continuous factors. This enables us to include all data, and to assess all effects in one go, without the need to set aside factors or to assess them separately in subsets of the data. This new analysis, together with the increased size of the data set, has confirmed that there is a highly significant effect of group as well as an interaction between group and HPR, showing that the adapted thresholds differ between the control and noise-exposed group, and that this difference becomes larger with increasing stimulus intensity.

This would also allow the authors to estimate the magnitude of the difference in 'threshold shift' between control and NE, which the authors do not present in the current manuscript. An estimate of this magnitude is necessary to evaluate the model in Figure 5. It appears that a threshold shift difference of ~12dB is necessary to obtain the difference between control and NE responses according to the model. Yet, judging from Fig. 3g, the actual threshold shift difference between control and NE is ~5dB. This would argue against gain adaptation being preserved.

In order to understand the effects of gain reduction seen in the population rate-level functions in Fig. 4, it is crucial to consider the distribution of thresholds, and particularly the fraction of units with thresholds within the HPR as well as below and above the HPR. We have thus added the cumulative distribution functions of the adapted thresholds to figure 6, to illustrate the relation more clearly. As demonstrated by the model, the dependence of gain upon adapted threshold is highly non-linear, thresholds below and up to the HPR mean are associated with shallow RIFs, whereas thresholds above the HPR mean can lead to very steep RIFs for just a few dB difference. For the 80 dB HPRs, the median threshold of the control group is at the HPR mean, whereas for the noise-exposed group, it is just below the HPR. In absolute dB, this difference is not huge, but due to the nonlinearity of the effect, it translates into a substantial difference in the average RIF steepness. We hope that our new analysis of RIF steepness changes through gain adaptation, presented in the revised figure 6, will help illustrate this more clearly. It should be noted that the model is only intended as a qualitative illustration of the mechanism, and not as a quantitative analysis tool.

I do not think these concerns are just statistical details. As presented, it appears that both central claims of the paper rest on a significant threshold shift difference. I would therefore encourage the authors to quantitatively nail this down.

We hope that the reviewer will find that our new analyses, together with the addition of more data, have achieved exactly this.

*2) A central conceptual concern is how the authors interpret “gain adaptation”. The authors do not clearly define this term in the paper. I strongly suggest they do so in the Introduction. The closest they come to defining it is in Line 396 – gain adaptation is necessary to maintain a stable long-term firing rate. If the claim that noise exposed animals (NE) have lower threshold adaptation is true, then by having intact gain adaptation, there is lower information about sounds at high prevailing noise levels. However, the question arises whether the preservation of long-term firing rate is an intended result or an epiphenomenon. An alternate explanation may be that the function of ‘gain adaptation’ (or alternate terminology) is to preserve the dynamic range of neural responses. At high thresholds, high gain ensures that the entire range of neural response rates is used for encoding. In this interpretation, it is a *failure* of ‘gain adaptation’ that leads to a loss of dynamic range in high noise levels. This decouples ‘gain adaptation’ from threshold adaptation. Given the issues with threshold adaptation (above), this might be an attractive alternative hypothesis to consider and discuss. The point here is that there may be a more straightforward explanation for what the authors observe.*

The reviewer raises an interesting point about the potential purpose of a gain adaptation mechanism for information processing. This prompted us to revise our analysis of gain changes, where we now calculate relative steepness of the RIFs for the different HPRs, with the RIFs obtained for the 44(56) stimulus (our quietest stimulus ensemble) as the reference for the calculation for the relative steepness. Changes in steepness are then related to the location of the RIF threshold relative to the HPR mean. This analysis is presented in the revised figure 6. It illustrates very clearly (particularly for the 80 dB SPL HPR) that reductions in steepness can be seen in both the control and the noise-exposed group for MUCs with thresholds below the HPR mean, and increases in steepness for MUCs with higher thresholds (Fig. 6 d,h). The major effect of noise-exposure concerns how many MUCs fall in each category, which is governed by threshold adaptation. We thus conclude that the same mechanism is active in both groups, and that by acting normally (instead of failing), it is creating the differences in RIF steepness and thus FI.

To avoid potential confusion about gain adaptation, which may indeed be interpreted quite differently by different readers, we have now phrased the analysis simply in terms of RIF steepness. In the interpretation and discussion of the results, we then introduce “gain changes” as the underlying mechanism of “changes in RIF steepness”.

3) Related to above, the authors do not comment on the two-peaked FI curves observed in the control condition (Fig. 4a), but is conspicuously absent in the NE condition. Does the second peak reflect the effect of high-threshold fiber inputs? If so, are the observed differences between control and NE basically the presence or absence of high-threshold inputs? If this is the case, one role of high-threshold inputs might be to preserve dynamic range at loud levels.

We agree with the reviewer on this point. Indeed, hypotheses along these lines were contained in an earlier draft of the manuscript, but removed for the sake of brevity. In the revised manuscript, we have added a statement to the discussion section.

4) Is gain adaptation really preserved? Although the mean spike rates in NE is roughly constant over HPRs, the magnitude is reduced significantly (as the authors point out). In the model, then, the constant-rate curve will be shifted to the right. Is this not a change in gain adaptation?

The main reason for the difference in mean firing rates was a difference in average subthreshold (spontaneous) firing rates between exposed and control animals, which would correspond to an up/down shift of the RIFs, not a change in suprathreshold gain at this stage of processing. This would mean that the gain adaptation mechanism works in the same way, just on a different baseline. However, after adding the new data and applying stricter MUC selection criteria, the difference in mean rates turned out to be a spurious effect, see revised figure 5.

5) Did the authors attempt any spike sorting on their data, given that they used tetrodes? Are single-unit data consistent with the multi-unit data that are presented? While they observe population-level effects from the multi-unit data, it is disappointing that they do not mention how single units behave because this could provide insight into underlying mechanisms. For example, the observed two-peaked FI curves could arise from distinct single neurons, which could carry implications for how high-threshold fibers affect AN activity. Further analyses comparing sorted single-unit data to the multi-unit data presented would significantly strengthen the paper.

We agree with the reviewer that single unit data would have been the best option to answer questions about the adaptation mechanisms in the greatest possible detail. However, for reasons that we do not quite understand, we unfortunately did not succeed in isolating single units from the tetrode data, despite trying several approaches with different spike sorting packages including KlustaKwik. We thus chose to extract multi-unit clusters through thresholding. In the revised manuscript, this process is now described in more detail. The new data that we have added has been obtained using linear Neuronexus probes, which are best suited for recording multi units anyway. The reason for using these probes in the new experiments was that these experiments were part of a follow-up project for which we needed a good coverage of the whole CF range of the IC from each animal, which is best obtained with the linear probes. Thus, also our extended data set only contains MUCs. However, for the analyses that we have performed here, and especially for detailing the effects of noise exposure on the encoding of information about sound level by the neuronal population, which might be the most relevant feature, MUCs are sufficient to yield a complete picture. For other aspects like understanding the origin of the two-peaked FI distributions, MUCs also yield some insight, as the two-peaked distributions can to some extent be explained by MUCs with high thresholds (compare scatter plots in figure 3). While single units would have enabled a better quantitative treatment of the effect, as there were occasional MUCs with “stepped” RIFs which might have caused an underestimation of the proportion of units with high thresholds, the main effect of noise exposure, which was the primary focus of our study, is clearly visible in the MUC data.

Minor concerns:

1) The authors mention that they obtain FRAs. Could they comment on whether there were differences between control and NE FRAs at high sound levels?

This would have been an interesting analysis. However, due to time constraints in the experiments, we recorded FRAs with three repetitions only, and thus we chose not to do detailed analyses of the FRAs beyond estimation of CFs and thresholds, as the FRAs were not recorded with enough repetitions to warrant more detailed analyses. Moreover, a detailed analysis of FRA shapes would probably be most meaningful for single units.

2) Please provide details on the number of units recorded, inclusion criteria for analysis, and the number of units analyzed. Since recordings are from a tetrode array, and it seems that each channel was treated independently, how do you ensure that units are not replicated across channels in a given tetrode? Some of this information is in a figure legend, but it belongs in the Methods section.

We have amended the methods section accordingly and added the relevant unit numbers to the results section. We generally only picked the recording from one electrode contact as the representative MUC for a tetrode. In exceptional cases, where the FRAs showed a difference in CF of more than 0.5 octaves for the recordings from two electrode contacts of one tetrode, we included both. The main criterion for inclusion of MUCs in further analysis was a rate-level function with a clearly defined threshold, which was verified through visual inspection of all fit results. In the original manuscript, we also included MUCs which only gave a good response for some of the conditions, and only included them for these conditions, which led to a varying number of units in the different analyses. In the new data set, only MUCs with clearly defined thresholds for all stimulus conditions have been included, and thus the unit number is now constant across all analyses.

3) Were these units recorded from the core or shell of IC? This is an important distinction and should be stated in Methods.

The units were recorded from the core of the IC. We have added this important information to the methods section, thanks for pointing this out

L78: "... suprathreshold firing rates were lower", but L310 "...adapted firing rates were slightly higher than in control animals for the quieter HPRs". I think L78 refers to the overall mean, and not all suprathreshold firing. Please clarify.

These statements have been rewritten in the revised manuscript.

L213: Is the t-test paired? Were multiple t-tests run to test whether 1-day post and pre were different at each frequency? If so, there is a multiple comparisons issue. Possibly, the 48kHz data point will not survive multiple comparisons correction. Since the authors do have many more NE neurons at 48-64 Khz, I wonder if they are underestimating their effects by including neurons that were in a frequency range not affected by NE. It might be worthwhile to drop the 48-64KHz neurons on this basis.

As the animals in our second set of experiments were only ABR'd up to 32 kHz, we have removed 48 kHz from the ABR analysis. For the analysis of IC recordings, we have opted to keep MUCs from the whole CF range, as the distributions of CFs are well matched in the extended data set.

L223: Any anatomy to confirm synaptopathy with your noise exposure paradigm would be nice.

The cochleae were used for an electron microscopic study, where a detailed investigation of synaptopathy was carried out on representative inner hair cells, including 3-D reconstruction. The results of this analysis have been submitted for publication as a separate manuscript (Bullen et al., in revision), as they were much too comprehensive to be included in this paper. Additionally, we have

already published histology confirming synaptopathy in a previous study where we have used the exact same noise exposure paradigm and equipment (Hesse et al., 2016). Moreover, one might also refer to the detailed results from the Liberman lab, as our noise exposure paradigm was modelled on the one used by them to induce cochlear synaptopathy.

L520: This is an intriguing suggestion, and possibly easily tested using something as simple as earplugs. Is there any published data testing this that might support your suggestion?

As there is no published data on this, we are currently working on this, conducting pilot experiments in human subjects with normal hearing to assess the effects of presentation levels and attenuation through earplugs. Our plan is to extend the investigation to hearing-impaired subjects with only a slight noise-induced hearing loss, as they are probably the group of subjects with the highest probability of showing a significant effect of synaptopathy on hearing function.

Reviewers' Comments:

Reviewer #1:

Remarks to the Author:

The authors have addressed all my comments.

Reviewer #2:

Remarks to the Author:

The authors have addressed most of my earlier concerns. I believe that the overall message is much improved. Some of the writing should be clarified to ensure that this reaches a wide audience. But some of my original concerns were not completely addressed. I think that the remaining concerns can be addressed in a minor revision.

The authors have run into familiar technical issues faced by many labs, including our own, at isolating single units from these tetrode and linear probes. While I do think that this would have been valuable information at supporting a role for high-threshold ANFs in the observed phenomena, the MU data presented sufficiently support the authors central claims.

Their multi-unit criteria, now completely described, are in line with that used by other labs. I appreciate the effort of the authors to increase both data quantity and quality. The statistical treatment in Fig. 3 is much better, but a concern remains.

Minor concerns and suggestions:

1) The reason for bringing up the box-and-whisker plots in Fig. 3 was possibly misunderstood. A description of this standard plot (as now provided in Fig. 3's legend) is not necessary. Rather, the question is about the normality of the data in Fig. 3, and the appropriateness of using parametric statistical methods such as ANOVAs that assume normally distributed data. The threshold differences between control and noise-exposed mice driving the effect in Fig. 3 seem to arise from non-normal distributions (especially control black marginal histograms in 3c, 3e, 3f. A statement commenting on the appropriateness of their statistics should be included in the main text.

2) The support for the model from data is described in Fig. 6, but is based on "tendencies" and no statistical analysis is provided. Please provide statistics, and if not significant, state so with a description of the tests used.

3) The authors explanation of "preserved gain adaptation" remains counter-intuitive. On the one hand, they argue that gain adaptation is preserved, but on Page 22, Line 453 they state: "...noise damage to affect high-threshold ANFs more strongly (Furman et al., 2013), indicating that high-threshold ANFs are crucial for adaptation to loud acoustic environments, as their loss cannot be compensated through neural adaptation in the central auditory system". This seems contradictory.

3a) Related to this, please define threshold adaptation and gain adaptation in your Introduction. Page 4, Line 70, right after you introduce neurons adapting their response functions, is a good place to do this.

3b) On Page 19, Line 410, the authors state that "... gain adaptation appears actively to have reduced neural response". Why actively? If anything, this effect happens because of the loss of a response component (high-threshold ANFs).

- 4) How many multiunit clusters from control and noise-exposed mice are the analyses based upon?
- 5) Page 22, Line 460: I appreciate that the authors bring up the bimodality of the FI curves. This may be a good place to state something to the effect of "Further experiments using single-unit recordings may be necessary to verify that this is the case".
- 6) Page 12/Line 244 – The Chambers et al. (2016) paper cited by the authors, and a recent follow up (Asokan et al., 2018) are highly relevant to the discussion of central gain in the Results section, and should be cited there.
- 7) Page 21, Line 425 - "designed to damage selectively" to "designed to selectively damage"
- 8) Page 21, Line 448 - "several folds less information" is overstating the effect size which is about 2-fold. Please restate.
- 9) For ease of reading, please label Figs. 2e and f as "Control" and "Noise exposed".
- 10) In the rebuttal, in response to my question about whether recordings were from core IC, they state: "We have added this important information to the methods section...". I cannot find it added to the Methods section. This is an important distinction especially given recent results (Asokan et al., 2018).